# Comparisons of Vaccine Hesitancy across Five Low- and Middle-Income Countries

**DOI:** 10.3390/vaccines7040155

**Published:** 2019-10-18

**Authors:** Abram L. Wagner, Nina B. Masters, Gretchen J. Domek, Joseph L. Mathew, Xiaodong Sun, Edwin J. Asturias, Jia Ren, Zhuoying Huang, Ingrid L. Contreras-Roldan, Berhanu Gebremeskel, Matthew L. Boulton

**Affiliations:** 1Department of Epidemiology, School of Public Health, University of Michigan, 1415 Washington Heights, Ann Arbor, MI 48109, USA; mastersn@umich.edu (N.B.M.); bgebreme@umich.edu (B.G.); mboulton@umich.edu (M.L.B.); 2Department of Pediatrics, University of Colorado Anschutz Medical Campus, B065, 13123 E 16th Ave, Aurora, CO 80045, USA; Gretchen.domek@childrenscolorado.org (G.J.D.); Edwin.asturias@cuanschutz.edu (E.J.A.); 3Center for Global Health, Colorado School of Public Health, A090, 13199 E Montview Blvd, Suite 310, Aurora, CO 80045, USA; 4Advanced Pediatrics Centre, Post Graduate Institute of Medical Education and Research, Chandigarh 160012, India; dr.joseph.l.mathew@gmail.com; 5Department of Immunization Program, Shanghai Municipal Centers for Disease Control & Prevention, NO. 1380, West Zhongshan Road, Shanghai 200336, China; sunxiaodong@scdc.sh.cn (X.S.); renjia@scdc.sh.cn (J.R.); huangzhuoying@scdc.sh.cn (Z.H.); 6Department of Epidemiology, Colorado School of Public Health, B119, 13001 E 17th Place, Aurora, CO 80045, USA; 7Center for Health Studies, Universidad del Valle de Guatemala, 18 Av. 11-95, Zona 15, Vista Hermosa III, Guatemala City 01015, Guatemala; Icontreras@ces.uvg.edu.gt; 8Department of Internal Medicine, Division of Infectious Disease, University of Michigan Medical School, 1500 East Medical Center Drive, Ann Arbor, MI 48109, USA

**Keywords:** vaccines, vaccine hesitancy, low- and middle-income countries, China, Bangladesh, Ethiopia, Guatemala, India

## Abstract

Vaccine hesitancy is a continuum of behaviors ranging from delay in receipt to vaccination refusal. Prior studies have typically focused on high-income countries, where vaccine hesitancy is particularly prevalent in more affluent groups, but the relationship between socioeconomic status and vaccine hesitancy in Low- and Middle-Income Countries (LMICs) is less clear. The aim of this study was to describe vaccine hesitancy in five LMICs. Mothers of children in Sirajganj, Bangladesh (*n* = 60), Shanghai, China (*n* = 788), Addis Ababa, Ethiopia (*n* = 341), Guatemala City and Quetzaltenango, Guatemala (*n* = 767), and Chandigarh, India (*n* = 309), completed a survey between 2016 and 2018 using the WHO’s 10-item Vaccine Hesitancy Scale. The scores of different constructs were compared across countries and by the mother’s education level using linear regression models with generalized estimating equations. Compared to mothers in China, mothers in Bangladesh perceived less vaccination benefit (β: 0.56, P = 0.0001), however, mothers in Ethiopia (β: −0.54, P < 0.0001) and Guatemala (β: −0.74, P = 0.0004) perceived greater benefit. Education level was not significantly linked with vaccine hesitancy. Local circumstances are important to consider when developing programs to promote vaccines. We did not find consistent associations between education and vaccine hesitancy. More research is needed to understand socio-cultural influences on vaccine decision-making.

## 1. Introduction

Worldwide, the number of deaths due to vaccine-preventable diseases (VPDs) has decreased precipitously in the past few decades. Excluding tuberculosis and cancers, the number of deaths from vaccine-preventable diseases declined from 4.1 million in 1990 (9% of all deaths) to 2.7 million in 2010 (5% of all deaths) [1]. Between 2000 and 2016, there was an average annual decline of 3.7% in under-5 mortality throughout the world, with 38.4 deaths per 1000 live births in 2016 [2]. However, the under-5 mortality rate in low- and middle-income countries (LMICs) is much higher than in high income countries. The under-5 mortality rate is 11.8/1000 in China, 26.7/1000 in Guatemala, 34.0/1000 in Bangladesh, 39.2/1000 in India, and 43.4/1000 in Ethiopia compared to 5.4/1000 for high income countries [2]. Expanding robust, routine immunization programs in LMICs is one important way to reduce incidence of VPDs and lower childhood mortality.

Vaccine uptake can be thought of as the result of the 5 As: Access, affordability, awareness, activation (e.g., cues such as SMS reminders), and acceptance [3]. While the relative contribution of each ‘a’ to vaccine uptake differs across countries, this last term, acceptance, or its inverse, hesitancy, has not been well studied in LMICs, despite its connection to global problems eradicating polio and eliminating neonatal tetanus in certain LMICs [4]. Vaccine hesitancy is also posing a threat to measles eradication goals [5]. Though the Americas originally declared measles eliminated in 2002, declining coverage of measles-mumps-rubella (MMR) vaccine and growing vaccine hesitancy in Brazil—particularly doubts about vaccine safety and effectiveness [6]—was accompanied by endemic transmission of measles over a year between 2013 and 2015. Measles was re-declared to be eliminated in the Americas in 2016, [7] but challenges persist through the region that threaten this elimination status again. For example, outbreaks of measles in 2018 and 2019 across the United States have been concentrated in ethnic groups or counties with low vaccination coverage and high rates of nonmedical vaccination exemptions (NMEs) [8]. All regional offices of the World Health Organization (WHO) have now planned to eliminate measles by 2020 [9], but this eradication goal is likely to be missed given sub-optimal vaccination coverage and persistent vaccine hesitancy in certain groups. Given the rising tide of vaccine hesitancy across the globe, the WHO identified vaccine hesitancy as one of the top ten global health threats in 2019 [10].

The World Health Organization (WHO) Strategic Advisory Group of Experts on Immunization (SAGE) put forward the following definition of vaccine hesitancy: “Vaccine hesitancy refers to delay in acceptance or refusal of vaccines despite availability of vaccination services. Vaccine hesitancy is complex and context specific, varying across time, place, and vaccines. It is influenced by factors such as complacency, convenience, and confidence” [11]. The WHO SAGE also developed a 10-item vaccine hesitancy scale to assess vaccine hesitancy and vaccine perceptions. [12] This has been implemented in Canada [13] and several LMICs—including Guatemala [14], Ethiopia [15], and China [16]. These studies have revealed interesting trends in the data—notably that a large proportion of the population believes in the effectiveness of vaccines, in general, while harboring some concerns about safety and the utility of certain vaccines. However, there exists a greater need to pool data from multiple countries together and examine global trends, as well as within- and between-country differences in vaccine hesitancy.

Another important consideration is how the relationship of socioeconomic status (SES) varies with vaccine hesitancy, and whether this is context-specific. Low vaccine uptake in LMICs has mostly been associated with individuals of lower SES [17,18,19,20,21], although this has been tied to disparities in access, affordability, and awareness, and less so on acceptance or hesitancy [22,23]. In contrast, more affluent individuals in high-income countries appear to be more vaccine hesitant and may have lower vaccine uptake [24,25,26]. The Wellcome Global Monitor 2018 also found that while 95% of surveyed individuals in South Asia and 92% of those in Eastern Africa believed that vaccines were safe, this figure was considerably lower in North America and Northern Europe (72% and 73%, respectively), with concerningly low perceptions of vaccine safety in Western Europe (59%) and Eastern Europe (50%) [27]. Overall, there is little research on vaccine hesitancy and SES in LMICs, although the findings may be mixed or show little relationship; for example a study in Shanghai found higher income lightly related to concerns about vaccine co-administration [28]. More information is needed on trends of vaccine hesitancy in LMICs.

Using previously conducted studies from five LMICs—including published studies from Guatemala [14], Ethiopia [15], and China [16], along with unpublished data from India and Bangladesh, this study aims to describe vaccine hesitancy in five LMICs using the WHO SAGE Vaccine Hesitancy Scale and compare country-specific vaccine hesitancy by education level.

## 2. Materials and Methods

The study sites were not systematically chosen but instead are a convenience sample of diverse settings in emerging economies, as shown in Table 1).

### 2.1. Bangladesh Study Population

In Siraganj, Bangladesh, mothers were selected from a convenience sample of individuals who were attending health clinics with a child in the eligible age range (<18 months). Six health clinics were chosen as a convenience sample from a list of clinics in the area based on their availability to have a data collector come regularly for interviews. Participants were verbally administered the questionnaire in Bengali.

### 2.2. China Study Population

In Shanghai, China, 40 township clinics were selected as part of a stratified, two-stage cluster sample. The first stage was at the township level with city districts forming a sampling strata, such that at least one township was selected from 15 districts (all districts in the municipality with the exception of Chongming Island). Within each township clinic, study staff initially selected a convenience sample of 30 caregivers with children under 3 months, in which fathers were excluded from the analysis in this study. Participants filled out a paper questionnaire in Mandarin Chinese.

### 2.3. Ethiopia Study Population

In Addis Ababa, Ethiopia, five of the ten sub-cities in Addis Ababa were selected as clusters, with one health center selected from each sub-city. Subjects at each health center were then systematically sampled with sampling proportionate to the size of the population served by each health center. Participants were eligible if they presented at the health center with a child aged 3–12 months for immunizations during the survey period. Data collectors were trained nurses at each health center. Participants were verbally administered the questionnaire in Amharic.

### 2.4. Guatemala Study Population

In Guatemala City and Quetzaltenango, Guatemala, participants were included if their child was between 6 weeks and 6 months of age and presenting for their first wellness visit at one of five public health clinics of the Ministry of Public Health and Social Assistance. The government clinics included in this study serve a low-income population with two of the clinics located in an urban setting surrounding Guatemala City and three in the rural southwest region of the country (Quetzaltenango). A study nurse assigned to each clinic verbally administered the surveys to participants using Spanish.

### 2.5. India Study Population

In Chandigarh, India, 30 Anganwadis (public child care locations) were chosen as the unit of cluster, randomly selected from a line list of all Anganwadis in Chandigarh. Data collectors were accompanied by Anganwadi workers and selected families who met the eligibility criteria based on a systematic selection within the community, with children under the age of 5 years. Participants were verbally administered the questionnaire in Hindi.

### 2.6. Vaccine Hesitancy Questionnaire and Survey

Items in the WHO SAGE 10-item Vaccine Hesitancy Scale were grouped based on internal consistency: Seven of the items corresponded to a construct relating to “lack of vaccination benefits” (standardized Crohnbach α = 0.86); the three other items were not internally consistent and are analyzed separately (“new vaccines risky”, “concerns about side effects”, and “some vaccines no longer needed”). The scores of different constructs were compared across country and by the mother’s education level using a linear regression model with generalized estimating equations (GEE). The outcome is a vaccine hesitancy construct on a scale from 1 to 5, with higher numbers indicating greater hesitancy in a given dimension, though this required reverse coding of questions L1, L2, L3, L4, L6, L7, and L8, as seen in Table 2. The correlation across items was assessed using a Pearson’s correlation coefficient.

### 2.7. Statistical Analysis

The distribution of vaccine hesitancy scores across these five LMICs was presented using descriptive statistics including means and standard deviations (sd). Multivariate models accounting for clustering by country were conducted using GEE models, (PROC GENMOD with a REPEATED statement), with clustering at both the country and sub-national survey-cluster accounted for. Analyses were weighted by the inverse of the number of participants in each survey so that each country’s survey generated the same weight for overall statistics, regardless of the size of the study population. Procedures were used in GEE models to account for the complex survey design to generate correct confidence intervals, and *p*-values for education and country-related differences in vaccine hesitancy scores. Significance was assessed at an α level of 0.05, and all analyses were conducted in SAS version 9.4 (SAS Institute, Cary, NC, USA).

### 2.8. Ethical Approval

This study was approved by institutional review boards or ethical review committees at the University of Michigan (#HUM00145902, #HUM00087564, #HUM00126383, and #HUM00126619), the Shanghai CDC (#2017-2), St. Paul’s Hospital Millennium Medical (#PM23/237), the Postgraduate Institute of Medical Education and Research (#PGI/IEC/2015/1363), the Colorado Multiple Institutional Review Board (#14-2170), Universidad del Valle Ethics Committee (#113-01-2015), and Guatemala National Ethics Committee of the Ministry of Public Health and Social Assistance (#11-2015).

## 3. Results

### 3.1. Demographics and Survey Details

Mothers of children in Sirajganj, Bangladesh (*n* = 60), Shanghai, China (*n* = 788), Addis Ababa, Ethiopia (*n* = 341), Guatemala City and Quetzaltenango, Guatemala (*n* = 767), and Chandigarh, India (*n* = 309), completed a survey between 2016 and 2018. The details about the country-specific surveys are found in Table 1. Education status varied between countries, with Bangladesh having 80% of the surveyed population with only a middle school education or less, though the population surveyed in China was far more educated, with 70% having at least some tertiary education (beyond high school). Ethiopia and Guatemala had 55% and 60% of the surveyed population with less than a middle school education, while the Indian survey showed 72% of participants had less than a middle school education, making China unique in this sample in terms of the distribution of educated individuals.

### 3.2. Vaccine Hesitancy Scale

The questions included from the WHO SAGE Vaccine Hesitancy Scale, L1–L10, are detailed in Table 2. All questions were asked on a Likert-scaled response system: Strongly Disagree (1), Disagree (2), Unsure (3), Agree (4), and Strongly Agree (5). As such, questions L1–4, L6, L7, and L8 were reverse-coded to assess between-item correlations, as shown in Table 3, and to create a hesitancy sum score with higher values corresponding to more hesitancy. Pearson correlation coefficients of the correlation between the different questions on the hesitancy scale revealed that while the positively worded items correlated moderately well with each other (L1–L4, L6, L7 all having correlation coefficients between 0.44–0.07), the positively worded questions did not correlate well with the negatively worded questions, even when reverse coded. The correlation of L5 to L1–L4 were all between −0.1 and 0.01, and L9′s correlations to the positively worded questions ranged from −0.22 to 0.07, while L10 showed low positive correlation with all questions.

### 3.3. Overview of Vaccine Hesitancy

Among all surveyed participants in Bangladesh, China, Ethiopia, Guatemala, and India, a large majority agreed or strongly agreed that “childhood vaccines are important for my child’s health” (95%), that “childhood vaccines are effective” (93%), and that “getting vaccines is a good way to protect my child” (94%). Around 90% of responders agreed or strongly agreed that “all vaccines offered by the government are beneficial”, “having my child vaccinated is important for others”, and “the information I receive about vaccines is trustworthy” (Figure 1). However, fewer respondents agreed that they follow their doctor’s recommendations about vaccines (79%). Among these positively-worded questions, the responses were generally more accepting, and less hesitant, than for the three negatively worded questions: L5, L9, and L10. The most predominant aspect of vaccine hesitancy was around risks of vaccines: With 62% of the surveyed population either disagreeing or unsure of whether “new vaccines carry more risks than older vaccines”, and 57% either disagreeing or unsure that they were “concerned about serious adverse effects of vaccines”.

Questions L1–L4 and L6–L8 represent the same overall construct, the benefit of vaccination. Thus when reverse-coded, they represented the lack of vaccination benefits, which, averaged across the five surveyed countries, had an overall score of 1.70 (sd 0.06), which was highest in Bangladesh (2.31, sd 0.11) and lowest in Guatemala (1.33, sd 0.21), as seen in Table 4. The fifth survey question pertained to the construct that new vaccines are riskier than older vaccines, which was less variable between countries, but China was found to have the most hesitancy regarding riskiness of new vaccines (3.53, sd 0.03). The highest vaccine hesitancy scores overall (3.12, sd 0.14) were about concern of side effects, where China and Guatemala had the highest hesitancy scores (3.83, sd 0.03 and 3.80, sd 0.29, respectively). Finally, the construct that some vaccines are no longer needed (those for which diseases are no longer common), had the most pronounced hesitancy scores in India (3.01, sd 0.06).

### 3.4. Multivariable Models

In the multivariable regression models (Table 5), compared to mothers in China, mothers in Bangladesh perceived less vaccination benefit (β: 0.56, P = 0.0001), but mothers in Ethiopia (β: −0.54, P < 0.0001) and Guatemala (β: −0.74, P = 0.0004) perceived greater benefit. Compared to Chinese mothers, mothers in Bangladesh (β: −0.65, P < 0.0001), India (β: −0.83, P < 0.0001), Guatemala (β: −0.97, P < 0.0001), and Ethiopia (β: −1.06, P = 0.0003) thought new vaccines less risky and, in three of these countries, had less concern about side effects (India β: −0.98, P < 0.0001; Bangladesh β: −1.05, P = 0.0087; and Ethiopia β: −1.68, P = 0.0064). Belief that some vaccines are no longer needed was less common in Ethiopia (β: −0.82, P < 0.0001) and Guatemala (β: −0.62, P < 0.0001), and more common in Bangladesh (β: 0.29, though not significant) and India (β = 0.35, P < 0.0001) compared to China. Education level was not significantly linked with vaccine hesitancy.

## 4. Discussion

Vaccine hesitancy has recently become an important area of scientific inquiry within high income countries as vaccine-preventable diseases, such as measles, have re-emerged across the United States and Europe. In contrast, far less is known about the role of vaccine hesitancy in LMICs. Due to the resurgence of vaccine-preventable diseases and plummeting vaccination rates fueled by vaccine hesitancy, the WHO identified vaccine hesitancy as one of the top ten global health threats in 2019 [10], and vaccine hesitancy is one of the reasons identified for the Global Vaccine Action Plan to miss its goals by 2020 [29]. Responding to the limited information available from LMICs (including middle-income countries, which face particular challenges implementing health programming given diminishing developmental aid [30]), we implemented vaccine hesitancy surveys in five sites throughout middle income countries in Latin America, Africa, and Asia. Our findings reveal substantial support for vaccines, with concerns about adverse events following immunization, the perceived relative riskiness of newer vaccines, and the continued use of older vaccines. We also did not find a significant impact of education level on vaccine hesitancy.

According to the research done by the Wellcome Trust in the Wellcome Monitor 2018, LMICs, in general, had lower rates of vaccine hesitancy and, for example, had fewer safety concerns about vaccines compared to HICs. The most hesitant countries were found in eastern Europe, and secondarily in western Europe and northern America. Bangladesh was one of the countries with the highest rates of agreement that vaccines are safe (97%), effective (97%), and that it is important for children to receive them (99%) [27]. Similarly, Ethiopia had 96% agreement that vaccines are safe, 97% agreement that they are effective, and 100% agreement that they are important for children to have. India also had high rates of agreement for all three axes: (95–98%) [27]. The Wellcome Trust survey did not ask about specific safety concerns. About half of mothers at our sites in Bangladesh, India, and Ethiopia expressed concerns about adverse events. Other surveys from multiple countries have revealed that the specific concerns that vaccine hesitant parents have for refusing vaccines can vary widely between sites [4]. The differences we saw between our sites could be due to background education levels in the population, exposure to different social and traditional media, and trust in the government, for example. Medical and political populism has also been linked to changes in vaccination attitudes [31,32]. Although we found no relationship between education and vaccine hesitancy, higher socioeconomic status has been linked to vaccine hesitancy in some previous studies [24,25,26].

We found a substantial proportion of individuals being relatively hesitant about newer vaccines. Messaging surrounding the introduction of new vaccines becomes very important—for instance, individuals are more likely to support new vaccines that protect against serious and fatal afflictions [33], and according to the affect heuristic, if an individual perceives more benefit to something, they will also believe it to be safer [34].

Worryingly, mothers expressed concerns about the continued use of vaccines that they supposed no longer necessary. Vaccines benefit the world through direct and indirect protection, and highly morbid diseases like measles and pertussis are quite contagious, and require a high level of vaccination coverage within a community to eliminate indigenous cases of disease [35,36]. Generations of parents have little first-hand experience with these diseases and so may judge their vaccinations to be unnecessary.

Although a discussion of vaccine hesitancy has been renewed in recent years because of outbreaks of vaccine-preventable disease in populations that should have easy access to vaccines [37], it has long been recognized as a potential problem [38]. There are no easy solutions to alleviating vaccine hesitancy [39]. Since doctors are seen as important sources of health information throughout the world, it will be key to give doctors the tools and education needed to deal with vaccine hesitancy, for instance, through nudges or talking about experiences with patients who had vaccine-preventable diseases [40].

### 4.1. The WHO SAGE Vaccine Hesitancy Scale

All five sites used the WHO SAGE Vaccine Hesitancy Scale [12]. Other instruments measuring vaccine hesitancy are also available including the Parent Attitudes about Childhood Vaccines (PACV) survey [41]. The PACV survey gives instructions on how to convert to and interpret an overall vaccine hesitancy score, whereas we came up with ad hoc measures in the WHO SAGE Vaccine Hesitancy Scale. We note several limitations of using this scale, including items L5, L9, and L10 not correlating well with the other items (perhaps as a result of their questions having reversed directionality relative to the other questions) [14,16]. Although this survey was simple to administer and translate, future research should identify how individuals in different regions of the world with different educational backgrounds interpret a 5-point Likert scale. Other scales are available, and the Parent Attitudes about Childhood Vaccines (PACV) has been particularly widely used [41,42,43]. PACV reports three dimensions of vaccine hesitancy—safety and efficacy, general attitudes, and behavior [44], which may better reflect the complexity of vaccination attitudes relative to the survey used in this article.

### 4.2. Strengths and Limitations

We were limited in our interpretations due to data collection issues across the different study sites, such as different dates and years of the survey, potential issues around the accuracy of translation and the local context, which may have altered the way in which translated text was interpreted, different modes of data collection at each study site, different uses of clusters between study sites, use of hospital vs. community-based samples, and variable ages of children across the five studies. A child’s age could affect a mother’s vaccine hesitancy through several mechanisms: Younger children receive more vaccines and more opportunities for side effects. As the collection of information on study participant characteristics was not uniform across datasets, we were unable to evaluate many individual-level predictors, which could be important contributors for vaccine hesitancy. However, this study does have a major strength of including children from multiple countries into one dataset, providing a more comprehensive data analysis with greater power, and being able to directly explore how much of the variability in vaccine hesitancy was due to between country differences or within country factors.

## 5. Conclusions

Local circumstances are important to consider when developing programs to promote vaccines, as thoughts about different aspects of vaccination are not uniform across countries. We did not find consistent associations about education and vaccine hesitancy, in contrast to prevailing findings about this relationship in high income countries. More work needs to be done on fully understanding socio-cultural influences on vaccine decision-making. Continued surveillance of attitudes towards vaccination in LMICs can help identify shifts in future opinions on vaccination attitudes.

## Figures and Tables

**Figure 1 vaccines-07-00155-f001:**
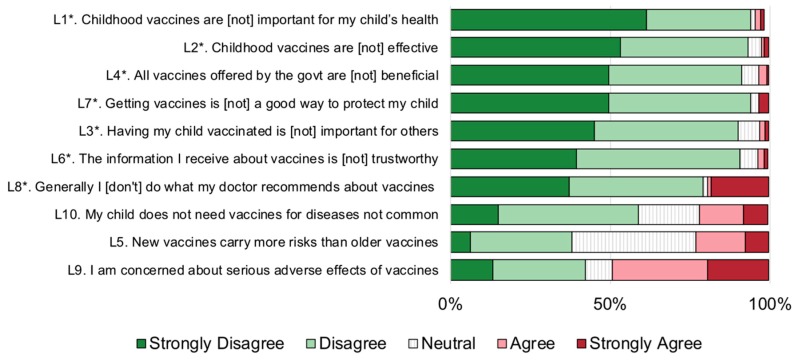
Overall patterns of vaccine hesitancy across the five surveyed Low- and Middle-Income Countries (LMICs), presented by question. Note, questions with a * were reverse coded so that the scale was in the same direction for all questions, with higher scores (1–5, 5 representing ‘Strongly Agree’, corresponding to higher vaccine hesitancy scores).

**Table 1 vaccines-07-00155-t001:** Survey details and educational attainment by site.

	Survey Details				Education Status			
Site	Dates	Cluster Location	Age of Children	n	N/A	≤ Middle School	High School	≥ Some Tertiary
Bangladesh	7/18–8/18	Health facility	<18 months	60	0	80%	18%	2%
China	5/17–9/17	Vaccination clinic	≤3 months	788	5	16%	13%	70%
Ethiopia	6/17–7/17	Vaccination clinic	3–12 months	341	3	55%	28%	16%
Guatemala	3/16–11/16	Health facility	6 weeks–6 months	767	0	60%	31%	9%
India	6/17–6/18	Anganwadi	≤5 years	309	0	72%	22%	6%

**Table 2 vaccines-07-00155-t002:** Vaccine hesitancy questions from World Health Organization (WHO) Strategic Advisory Group of Experts (SAGE) on immunization working group.

L1	Childhood Vaccines are Important for My Child’s Health
L2	Childhood vaccines are effective
L3	Having my child vaccinated is important for the health of others in my community
L4	All childhood vaccines offered by the government program are beneficial
L5	New vaccines carry more risks than older vaccines
L6	The information I receive about vaccines is reliable and trustworthy
L7	Getting vaccines is a good way to protect my child/children from disease
L8	Generally I do what my doctor or health care provider recommends about vaccines
L9	I am concerned about serious adverse effects of vaccines
L10	My child does not need vaccines for diseases that are not common anymore

**Table 3 vaccines-07-00155-t003:** Between-item Pearson’s correlation of SAGE WHO Vaccine Hesitancy Scale. Green shades represent positive correlations; red, negative. Upper right cells are blank to eliminate redundancies.

	L1 *	L2 *	L3 *	L4 *	L5	L6 *	L7 *	L8 *	L9
**L2 ***	0.70								
**L3 ***	0.50	0.55							
**L4 ***	0.52	0.64	0.57						
**L5**	−0.10	0.01	−0.05	−0.02					
**L6 ***	0.44	0.51	0.45	0.54	−0.01				
**L7 ***	0.60	0.62	0.49	0.58	−0.01	0.60			
**L8 ***	0.30	0.34	0.18	0.16	0.02	0.18	0.38		
**L9**	0.00	0.05	0.04	0.07	0.22	0.03	0.05	−0.22	
**L10**	0.18	0.16	0.21	0.16	0.21	0.20	0.21	0.19	0.07

* Reverse coded according to the original range of the scale.

**Table 4 vaccines-07-00155-t004:** Overall vaccine hesitancy score, and proportion of mothers who agree, stratified by construct, across the five surveyed LMICs.

		Lack of Vaccination Benefits ^†^	New Vaccines Risky	Concern about Side Effects	Some Vaccines No Longer Needed
		L1*-L4*, L6*-L8*	L5	L9	L10
Overall	mean ± SD	1.70 ± 0.06	2.86 ± 0.09	3.12 ± 0.14	2.55 ± 0.06
	%	1.7%	22.8%	48.9%	21.2%
Bangladesh	mean ± SD	2.31 ± 0.11	2.88 ± 0.11	2.62 ± 0.41	2.90 ± 0.25
	%	8.3%	10.0%	36.7%	30.0%
China	mean ± SD	1.58 ± 0.02	3.53 ± 0.03	3.83 ± 0.03	2.67 ± 0.05
	%	0.3%	44.7%	71.4%	26.8%
Ethiopia	mean ± SD	1.64 ± 0.10	2.57 ± 0.33	2.57 ± 0.29	2.06 ± 0.16
	%	0.0%	24.9%	25.5%	12.0%
Guatemala	mean ± SD	1.33 ± 0.21	2.58 ± 0.18	3.80 ± 0.29	2.11 ± 0.04
	%	0.0%	14.2%	71.7%	4.2%
India	mean ± SD	1.64 ± 0.04	2.75 ± 0.05	2.79 ± 0.07	3.01 ± 0.06
	%	0.0%	20.4%	39.2%	33.0%

* Reverse coded according to the original range of the scale. ^†^ For items in lack of vaccination benefits, standardized coefficient alpha = 0.86.

**Table 5 vaccines-07-00155-t005:** Results of linear models with generalized estimating equations for vaccine hesitancy constructs, clustering by country as well as sub-national clustering from each survey’s study design.

		Lack of Vaccination Benefits		New Vaccines Are Risky		Concern about Side Effects		Some Vaccines No Longer Needed	
		β	P	β	P	β	P	β	P
**Site**	**Bangladesh**	0.56	0.0001	−0.65	<.0001	−1.05	0.0087	0.29	0.1747
	**China**	ref	--	ref	--	ref	--	ref	--
	**Ethiopia**	−0.54	<0.0001	−1.06	0.0003	−1.68	0.0064	−0.82	<0.0001
	**Guatemala**	−0.74	0.0004	−0.97	<0.0001	−0.67	0.3263	−0.62	<0.0001
	**India**	−0.04	0.7008	−0.83	<0.0001	−0.98	<.0001	0.35	<0.0001
**Education**	**≤Middle School**	−0.01	0.7476	0.09	0.2142	−0.16	0.1240	0.02	0.7553
	**High School**	0.09	0.1571	−0.01	0.8676	−0.14	0.1136	−0.04	0.7078
	**≥College**	ref	--	ref	--	ref	--	ref	--

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
