# Peer review of "Comparisons of Vaccine Hesitancy across Five Low- and Middle-Income Countries"

_vaccines, 2019, doi:10.3390/vaccines7040155_

Round 1

Reviewer 1 Report

The article addresses the important issue of vaccine hesitancy in developing and low- and middle-income countries. Paying attention to developing countries in the context of vaccination is a good choice. Data and the methodology used are described is a clear and transparent manner. However, several issues should be explained:

- There is no clear justification why these five countries have been chosen for analysis. The authors mention that the largest percentage of people sceptical of vaccinations is in Eastern European countries (there is also a radical decrease in vaccinations in Ukraine) – the question is why developing countries in this region of the world are not taken into account?

- While numbers in statistical analyses always show certain trends dominant in a given population, numbers in figures showing research results always rise the question about the reasons for these and no other trends in replies. In other words, statistical data must always undergo proper interpretation. This is the weakest point of this article: there is no extensive part explaining and interpreting the results obtained.

- The authors assume that the country of origin and its economic development are indicators that may differentiate the attitude towards vaccines. They do not even mention other possible reasons, such as the atmosphere and influences of populism in health policy (Lasco, G, & Curato, N. (2019). Medical populism. Social Science & Medicine, 221, 1–8. https://doi.org/10.1016/j.socscimed.2018.12.006.  ), as well as cultural and political reasons (Piotr Å»uk, PaweÅ‚ Å»uk, Justyna Lisiewicz-Jakubaszko, The anti-vaccine movement in Poland: The socio-cultural conditions of the opposition to vaccination and threats to public health, https://www.sciencedirect.com/science/article/pii/S0264410X19301586). These aspects are worth mentioning in the discussion and conclusions.

- You can also briefly describe what actions the authors suggest to reduce the vaccine hesitancy phenomenon. (You can use: Poland, G. A., Jacobson, R. M. (2001). Understanding those who do not understand: a brief review of the anti-vaccine movement. Vaccine, 19(17–19), 2440–2445. https://doi.org/10.1016/S0264-410X(00)00469-2.)

- Research samples in individual countries do not need to be homogeneous – respondents differ not only between countries, but also within individual populations. Therefore, it is also worth highlighting in the article the directions and framework of future research on the attitude towards vaccination: e.g. showing that it is not only about the respondents’ country of origin, but also about their social position in a given society, education rate, place of residence (rural areas – a city) and the assessment of the quality of the health-care system in the country. These variables may be more significant than the country of origin.

Author Response

The article addresses the important issue of vaccine hesitancy in developing and low- and middle-income countries. Paying attention to developing countries in the context of vaccination is a good choice. Data and the methodology used are described is a clear and transparent manner. However, several issues should be explained:

- There is no clear justification why these five countries have been chosen for analysis. The authors mention that the largest percentage of people sceptical of vaccinations is in Eastern European countries (there is also a radical decrease in vaccinations in Ukraine) – the question is why developing countries in this region of the world are not taken into account?

Authors’ response:  The countries were not systematically chosen and we agree that other countries would be interesting to look at, which is why in the discussion we discuss limitations in comparisons between countries.

In the methods we now write (line 102): “The study sites were not systematically chosen but instead are a convenience sample of diverse settings in emerging economies. “

-------------

- While numbers in statistical analyses always show certain trends dominant in a given population, numbers in figures showing research results always rise the question about the reasons for these and no other trends in replies. In other words, statistical data must always undergo proper interpretation. This is the weakest point of this article: there is no extensive part explaining and interpreting the results obtained.

Authors’ response:   We have edited the manuscript to make sure that there is a threat from the methods, through the results, to the conclusions. For instance, we now explain in the methods: “The correlation across items was assessed using a Pearson’s correlation coefficient.” We had already discussed the results of this in section 3.2, and in the discussion in section 4.1. 

We now explain better in the discussion some descriptive analyses (line 153, section 2.7): “The distribution of vaccine hesitancy scores across these five LMICs was presented using descriptive statistics including means and standard deviations (sd).” These results are presented in section 3.4 and discussed in second paragraph of discussion (around line 268).

In the results we present a model with education as a predictor. We now add a sentence in the discussion to this point (line 271): “Although we found no relationship between education and vaccine hesitancy, higher socioeconomic status has been linked to vaccine hesitancy in some previous studies [24–26].”

-------------

- The authors assume that the country of origin and its economic development are indicators that may differentiate the attitude towards vaccines. They do not even mention other possible reasons, such as the atmosphere and influences of populism in health policy (Lasco, G, & Curato, N. (2019). Medical populism. Social Science & Medicine, 221, 1–8. https://doi.org/10.1016/j.socscimed.2018.12.006.  ), as well as cultural and political reasons (Piotr Å»uk, PaweÅ‚ Å»uk, Justyna Lisiewicz-Jakubaszko, The anti-vaccine movement in Poland: The socio-cultural conditions of the opposition to vaccination and threats to public health, https://www.sciencedirect.com/science/article/pii/S0264410X19301586). These aspects are worth mentioning in the discussion and conclusions.

Authors’ response:   We have now cited this papers in the discussion (line 270): “Medical and political populism has also been linked to changes in vaccination attitudes [31,32]. “ 

-------------

- You can also briefly describe what actions the authors suggest to reduce the vaccine hesitancy phenomenon. (You can use: Poland, G. A., Jacobson, R. M. (2001). Understanding those who do not understand: a brief review of the anti-vaccine movement. Vaccine, 19(17–19), 2440–2445. https://doi.org/10.1016/S0264-410X(00)00469-2.)

Authors’ response:  We appreciate the suggestion of referencing a very pertinent paper. We include it in a new paragraph in the discussion (line 319):

“Although a discussion of vaccine hesitancy has been renewed in recent years because of outbreaks of vaccine-preventable disease in populations that should have easy access to vaccines [37], it has long been recognized as a potential problem [38]. There are no easy solutions to alleviating vaccine hesitancy [39]. Since doctors are seen as important sources of health information throughout the world, it will be key to give doctors the tools and education needed to deal with vaccine hesitant, for instance through nudges or talking about experiences with patients who had vaccine-preventable disease [40].”

-------------

- Research samples in individual countries do not need to be homogeneous – respondents differ not only between countries, but also within individual populations. Therefore, it is also worth highlighting in the article the directions and framework of future research on the attitude towards vaccination: e.g. showing that it is not only about the respondents’ country of origin, but also about their social position in a given society, education rate, place of residence (rural areas – a city) and the assessment of the quality of the health-care system in the country. These variables may be more significant than the country of origin.

Authors’ response:   This is an important limitation that we now list in our discussion (line 310): “Because the collection of information on study participant characteristic was not uniform across datasets, we were unable to evaluate many individual-level predictors, which could be important contributors for vaccine hesitancy. “ 

-------------

Reviewer 2 Report

This study compares vaccine hesitancy across five LMICs, using the WHO SAGE 10-item Vaccine Hesitancy Scale. The authors acknowledge the limitations of interpretations given the differences in modes of data collection, sampling strategies, age ranges of children etc used across the 5 countries. Despite these limitations, the study makes a valuable contribution by directly examining variability of vaccine hesitancy across 5 LMICs where much of the extant literature has focused on HICs.

Lines 49-54: How much of these childhood mortality rates are attributable to VPDs?

Move Table 2 (survey details section) to “Materials and Methods” where you describe the different study populations.

The authors should provide a clear rationale for using the WHO SAGE scale for this study (over other commonly used vaccine hesitancy scales such as PACV) and address strengths and weaknesses of the SAGE scale. For example, 10 5-point Likert scale items in SAGE vaccine hesitancy scale seems inattentive to the conceptual dimensionality of “vaccine hesitancy.” Further, only three of the ten items are reverse-worded, raising concerns about an acquiescence effect increasing measurement error. What, if any, are other specific measurement and/or survey design limitations of the SAGE vaccine hesitancy scale employed? You could expand on this in section 4.1.

Line 180: This should be Table 1, not Table 2. Please move Tables and figures nearer to where they are discussed in the main text.

Section 3.3: Do these percentages refer to the aggregate sample?

What is the rationale for selecting China as the reference category/country? Related to this, on what basis were these 5 LIMCs selected for this study?

Author Response

This study compares vaccine hesitancy across five LMICs, using the WHO SAGE 10-item Vaccine Hesitancy Scale. The authors acknowledge the limitations of interpretations given the differences in modes of data collection, sampling strategies, age ranges of children etc used across the 5 countries. Despite these limitations, the study makes a valuable contribution by directly examining variability of vaccine hesitancy across 5 LMICs where much of the extant literature has focused on HICs.

Authors’ response:  We appreciate the reviewer’s comments.  

-------------

Lines 49-54: How much of these childhood mortality rates are attributable to VPDs?

Authors’ response:  This is an interesting question and unfortunately we found no studies which directly linked the number of VPD deaths to the total number of deaths in children. Using the first reference we were able to calculate deaths in all age groups however (line 46): “Excluding tuberculosis and cancers, the number of deaths from vaccine-preventable diseases declined from 4.1 million in 1990 (9% of all deaths) to 2.7 million in 2010 (5% of all deaths) [1]” .  

-------------

Move Table 2 (survey details section) to “Materials and Methods” where you describe the different study populations.

Authors’ response: We moved Table 2 to methods.   

-------------

The authors should provide a clear rationale for using the WHO SAGE scale for this study (over other commonly used vaccine hesitancy scales such as PACV) and address strengths and weaknesses of the SAGE scale. For example, 10 5-point Likert scale items in SAGE vaccine hesitancy scale seems inattentive to the conceptual dimensionality of “vaccine hesitancy.” Further, only three of the ten items are reverse-worded, raising concerns about an acquiescence effect increasing measurement error. What, if any, are other specific measurement and/or survey design limitations of the SAGE vaccine hesitancy scale employed? You could expand on this in section 4.1.

Authors’ response:   Other studies have already detailed some weaknesses of the SAGE scale, so we do not think it appropriate to write all these points. We cite these now in line 288:

Domek GJ, O’Leary ST, Bull S, Bronsert M, Contreras-Roldan IL, Bolaños Ventura GA, et al. Measuring vaccine hesitancy: Field testing the WHO SAGE Working Group on Vaccine Hesitancy survey tool in Guatemala. Vaccine 2018:5273–81. doi:10.1016/j.vaccine.2018.07.046.

Ren J, Wagner AL, Zheng A, Sun X, Boulton ML, Huang Z, et al. The demographics of vaccine hesitancy in Shanghai, China. PLoS One 2018;13:e0209117. doi:10.1371/journal.pone.0209117.

We also now have a larger discussion section which includes the PACV (line 290):

“Other scales are available, and the Parents Attitude Towards Childhood Vaccines (PACV) has been particularly widely used [35–37]. PACV reports three dimensions of vaccine hesitancy – safety and efficacy, general attitudes, and behavior [38], which may better reflect the complexity of vaccination attitudes relative to the survey used in this article.”

-------------

Line 180: This should be Table 1, not Table 2. Please move Tables and figures nearer to where they are discussed in the main text.

Authors’ response:   We have moved Table 2 to be closer to the methods (now it is Table 1). The other tables /figures are pasted right after they are first referenced. 

-------------

Section 3.3: Do these percentages refer to the aggregate sample?

Authors’ response:  Yes. In line 196 we modified it to read “Among all surveyed participants.” And in line 155 in the methods we expanded a sentence to now read: “Analyses were weighted by the inverse of the number of participants in each survey so that each country’s survey generated the same weight for overall statistics, regardless of the size of the study population. “  

-------------

What is the rationale for selecting China as the reference category/country? Related to this, on what basis were these 5 LIMCs selected for this study?

Authors’ response:  In the methods we now write (line 102): “The study sites were not systematically chosen but instead are a convenience sample of diverse settings in emerging economies (Table 1). “

China was chosen as the reference category in the models because it had the largest sample size.

-------------